# Host syndecan-1 promotes listeriosis by inhibiting intravascular neutrophil extracellular traps

**Rafael S. Aquino[1], Atsuko Hayashida[1], Pyong Woo Park[1,2]\***

**1** Department of Medicine, Boston Children's Hospital, Boston, MA, United States of America, **2** Department of Pediatrics, Harvard Medical School, Boston, MA, United States of America

\* pyong.park@childrens.harvard.edu

**Data Availability Statement:** All relevant data are within the manuscript and its Supporting Information files.

**Funding:** This work was supported in part by National Institutes of Health Grant R01 HL142213

## Abstract

Heparan sulfate proteoglycans (HSPGs) are at the forefront of host-microbe interactions. Molecular and cell-based studies suggest that HSPG-pathogen interactions promote pathogenesis by facilitating microbial attachment and invasion of host cells. However, the specific identity of HSPGs, precise mechanisms by which HSPGs promote pathogenesis, and the *in vivo* relevance of HSPG-pathogen interactions remain to be determined. HSPGs also modulate host responses to tissue injury and inflammation, but functions of HSPGs other than facilitating microbial attachment and internalization are understudied in infectious disease. Here we examined the role of syndecan-1 (Sdc1), a major cell surface HSPG of epithelial cells, in mouse models of *Listeria monocytogenes* (*Lm*) infection. We show that *Sdc1-/-* mice are significantly less susceptible to both intragastric and intravenous *Lm* infection compared to wild type (Wt) mice. This phenotype is not seen in *Sdc3-/-* or *Sdc4-/-* mice, indicating that ablation of Sdc1 causes a specific gain of function that enables mice to resist listeriosis. However, Sdc1 does not support *Lm* attachment or invasion of host cells, indicating that Sdc1 does not promote pathogenesis as a cell surface *Lm* receptor. Instead, Sdc1 inhibits the clearance of *Lm* before the bacterium gains access to its intracellular niche. Large intravascular aggregates of neutrophils and neutrophil extracellular traps (NETs) embedded with antimicrobial compounds are formed in *Sdc1-/-* livers, which trap and kill *Lm*. *Lm* infection induces Sdc1 shedding from the surface of hepatocytes in Wt livers, which is directly associated with the decrease in size of intravascular aggregated NETs. Furthermore, administration of purified Sdc1 ectodomains or DNase inhibits the formation of intravascular aggregated neutrophils and NETs and significantly increases the liver bacterial burden in *Sdc1-/-* mice. These data indicate that *Lm* induces Sdc1 shedding to subvert the activity of Sdc1 ectodomains to inhibit its clearance by intravascular aggregated NETs.

## Author summary

Listeriosis is a rare but deadly infectious disease caused by *Lm*, a facultative intracellular Gram-positive bacterium. Immunocompromised individuals, pregnant women, and the

(PWP). The funder has no role in study design, data collection and analysis, decision to publish, or preparation of the manuscript.

**Competing interests:** The authors have declared that no competing interests exist.

very young and elderly are particularly at risk for serious *Lm* infections. *Lm* has adapted several ingenious mechanisms to subvert host cell biology to invade, hide, and survive in intracellular compartments. The intracellular virulence mechanisms of *Lm* have been extensively studied, but how the bacterium overcomes eradication prior to its intracellular life cycle remains largely unknown. A significant number of extracellular bacteria are present in tissues at early phases of pathogenesis, and several arms of extracellular innate immunity have been shown to be important in attenuating disease progression. However, precisely how and where innate immunity limits and clears *Lm* is incompletely understood. We discovered that neutrophils swarm around *Lm* in hepatic blood vessels and extrude DNA fibers embedded with antimicrobial compounds to kill *Lm*. Sdc1 promotes *Lm* pathogenesis by inhibiting this unique defense mechanism when shed from the cell surface of hepatocytes by *Lm* infection. Our results uncover a previously unknown subversion mechanism where pathogenic bacteria co-opt a host extracellular matrix (ECM) receptor to counteract innate immune defense.

## Introduction

Microbial attachment and invasion of host cells are necessary steps in the establishment of infection [1]. A large number of viruses, bacteria, and parasites bind to the heparan sulfate (HS) moiety of HSPGs, and these interactions have been shown to facilitate pathogen attachment and invasion of host cells *in vitro* [2–4]. HSPGs are complex glycoconjugates comprised of one or several HS chains attached covalently to specific core proteins [5]. HSPGs are expressed ubiquitously on the cell surface and in the ECM, and they bind specifically and noncovalently to heparin-binding molecules through their HS chains. In infectious disease, cell surface HSPGs are thought to function primarily as coreceptors for heparin-binding microbial pathogens by localizing and increasing the concentration of pathogens at the host cell surface, thereby facilitating the interaction of pathogens with their respective signaling receptors.

However, there are several major gaps in our understanding of HSPG-pathogen interactions. First, current understanding of the role of HSPGs in infection is based predominately on *in vitro* systems. These studies advocate that HSPGs promote infection as an attachment and internalization receptor, but there are surprisingly very few supportive examples of this idea *in vivo*. In fact, some studies have suggested that the ability of pathogens to interact with HSPGs may be a result of cell culture adaptation [2, 6]. Second, the identity of cell surface HSPGs that facilitate attachment and invasion is unknown. Moreover, whether a specific cell surface HSPG functions simply as the initial attachment site, a direct internalization receptor, or as part of an internalization receptor complex remains to be determined. The structural heterogeneity of HS also presents a major challenge for elucidating the structural specificity of HS in microbial pathogenesis. Attempts to elucidate the essential structural features have not been successful largely because methods to directly sequence long HS polysaccharides are not available and HS polysaccharides with defined sulfation patterns are difficult to obtain. In addition, although HSPGs are multi-functional molecules, most studies have so far focused on their role as an attachment/internalization receptor, and have largely overlooked the possibility that HSPGs may have other functions in microbial pathogenesis.

To address these gaps in our understanding of HSPG-pathogen interactions, we examined the role of syndecans in the pathogenesis of *Lm* infection. The syndecan family of type I transmembrane HSPGs, comprised of four members in mammals (Sdc1 through Sdc4), is the major source of cell surface HS. Two to three HS chains are attached distal to the plasma

membrane on syndecan core proteins. Syndecans have multiple functions on the cell surface and as soluble, intact HSPG ectodomains in the extracellular environment when released by ectodomain shedding [5, 7]. *Lm* is a food-borne pathogen that can cross the intestinal mucosal barrier, enter the circulation, and cause bacteremia and meningitis [8]. Invasive listeriosis is not common, but listeriosis is a significant infectious disease that has a much higher rate of hospitalization (up to 95%) and mortality (up to 30%) than those caused by most other food-borne pathogens [9, 10]. Immunocompromised individuals, pregnant women, and the very young and elderly are particularly at risk for serious *Lm* infections [9]. Furthermore, listeriosis is common in patients receiving biologics and immunosuppressive therapies [11, 12], which are increasingly used to treat several major chronic diseases.

Available data suggest that syndecans, and in particular Sdc1, may facilitate several key steps of *Lm* pathogenesis. First, *Lm* ActA binds to the HS component of HSPGs [13]. ActA is a virulence factor best known for its capacity to manipulate the actin cytoskeleton to allow bacterial migration within and between host cells [14]. How ActA binding to HSPGs modulates *Lm* pathogenesis is unknown, but the cytoplasmic domain of Sdc1 can associate with the actin cytoskeleton [15], suggesting that Sdc1 may regulate both extracellular and intracellular interactions of ActA through its HS chains and cytoplasmic domain, respectively. Second, *Lm* InlB also binds to HSPGs [16–18]. InlB is a virulence factor that binds to the hepatocyte growth factor receptor c-Met [19] and complement receptor C1qR [20]. C1qR mediates the uptake of *Lm* in professional phagocytes, whereas Met mediates bacterial internalization in non-phagocytic cells, such as hepatocytes, by inducing the mono-ubiquitination of Met and subsequent clathrin-dependent endocytosis [21]. Importantly, InlB binds tightly to heparin, a pharmaceutical analog of HS, with nanomolar affinity [18] and heparin can potentiate the activation of Met by InlB by 10-fold [16, 17]. Structural studies of InlB indicate that the leucine-rich repeat region of InlB binds to a Met domain that does not bind to HGF, and C-terminal regions of InlB induce heparin-mediated Met clustering, which is required for full Met activation [22]. The predominant HSPG of epithelial cells, including hepatocytes, is Sdc1, suggesting that cell surface Sdc1 may function as a coreceptor for InlB that potentiates Met signaling and facilitates *Lm* attachment and internalization. Furthermore, most tissues infected by *Lm* contain polarized cells, such as intestinal epithelial cells, hepatocytes, and brain endothelial cells. While the exact mechanism is not fully understood, *Lm* enters epithelial cells through the basolateral surface [23], and Sdc1 is expressed predominantly on the basolateral surface of epithelial cells [5].

Consistent with these observations suggesting a pro-pathogenic role for Sdc1 in *Lm* pathogenesis, Sdc1 ablation causes a gain of function in both intragastric (i.g.) and intravenous (i.v.) mouse models of *Lm* infection, where *Sdc1-/-* mice are significantly less susceptible to *Lm* infection compared to Wt mice. However, our studies unexpectedly show that cell surface Sdc1 is not a *Lm* attachment and invasion receptor in both phagocytic and non-phagocytic cells. Instead, our results reveal a new virulence mechanism of *Lm* where the bacterium subverts the ability of Sdc1 to inhibit bacterial killing by neutrophil extracellular traps (NETs) in hepatic intravascular compartments.

## Results

### Sdc1 ablation is a gain of function mutation in *Lm* infection

We initially hypothesized that if *Lm* exploits the syndecan family of HSPGs for its pathogenesis, then hosts without syndecans will respond differently to *Lm* infection. Sdc1 is predominantly expressed by epithelial cells and plasma cells, Sdc2 is expressed by endothelial cells and fibroblasts, Sdc3 is expressed by neural crest-derived cells, and Sdc4 is expressed by most cell types, albeit at levels much lower than that of other syndecans [5, 24]. However, the exact

proteoglycome is not known for any given cell type or tissue *in vivo*. We therefore determined the expression of syndecans in target tissues of *Lm* and found that while all four syndecans are expressed, Sdc1 is most abundantly expressed in the liver and spleen (S1 Fig).

We next examined the response of *Sdc1-/-*, *Sdc3-/-* and *Sdc4-/-* mice on the BL/6J background to *Lm* infection. Unchallenged syndecan null mice are healthy, viable, fertile and largely indistinguishable from their Wt littermates. However, *Sdc1-/-* and *Sdc4-/-* mice show striking pathological phenotypes when challenged with instigators of tissue injury [3, 5, 25], suggesting that each syndecan functions specifically in post-developmental processes. Wt and syndecan null mice were infected i.g. with $10^{10}$ cfu of *Lm* strain EGDe and the bacterial burden in the liver and spleen was measured at 24 h post-infection (pi). A large infectious dose was required for these experiments because mice on the BL/6 background are relatively resistant to i.g. *Lm* infection [26]. The bacterial burden in both the liver and spleen was significantly reduced by approximately 10-fold and 7-fold, respectively, in *Sdc1-/-* mice compared to Wt mice, whereas the cfu counts in *Sdc3-/-* and *Sdc4-/-* tissues were similar to those of Wt mice (Fig 1A). To determine whether this gain of function phenotype of *Sdc1-/-* mice is restricted to the BL/6J background, we compared the response of Wt and *Sdc1-/-* mice on the more susceptible BALB/c background. *Sdc1-/-* mice on BALB/c infected i.g. with $10^{9}$ cfu of *Lm* EGDe also showed significantly decreased liver and spleen bacterial burdens compared to Wt mice infected identically (Fig 1B). These data indicate that Sdc1 ablation enables mice to specifically resist i.g. *Lm* infection and suggest that Sdc1 is a critical host factor that promotes *Lm* pathogenesis.

## Sdc1 ablation inhibits the progression of *Lm* infection after intestinal invasion and dissemination

We next evaluated the time course of *Lm* infection in Wt and *Sdc1-/-* mice. All subsequent studies were performed with mice on the BALB/c background because we confirmed that BALB/c mice are more susceptible to *Lm* infection compared to BL/6 mice. Wt and *Sdc1-/-* mice were infected i.g. with *Lm* EGDe and the tissue bacterial burden was assessed at various times pi. The bacterial load progressively increased in Wt liver (Fig 2A) and spleen (S2 Fig). However, the bacterial burden in *Sdc1-/-* liver (Fig 2A) and spleen (S2 Fig) was significantly decreased at 24 and 48 h pi compared to Wt tissues. Interestingly, the tissue bacterial load was similar in Wt and *Sdc1-/-* mice at 12 h pi (Fig 2A and S2 Fig), suggesting that Sdc1 ablation affects *Lm* pathogenesis after bacterial infection of intestinal tissues. Consistent with this idea, there was no significant difference in the intestinal bacterial burden between Wt and *Sdc1-/-* mice at 5 and 24 h pi (Fig 2B). Furthermore, the liver bacterial burden was significantly lower in *Sdc1-/-* mice compared to Wt mice at 24 h pi when infected i.g. with a mouse-adapted *Lm* strain, HEL-921, that expresses InlA that can bind to mouse E-cadherin [27] (Fig 2C). These results indicate that inefficient colonization of intestinal tissues does not underlie the significantly reduced cfu counts in *Lm*-infected *Sdc1-/-* liver and spleen. The bacterial load in *Sdc1-/-* livers was also significantly decreased relative to Wt livers when infected with *Lm* 10403S (S3 Fig), indicating that the gain of function phenotype of *Sdc1-/-* mice is not *Lm* strain-dependent.

Corroborating findings from the i.g. infection studies, similar differences in *Lm* virulence were seen in Wt and *Sdc1-/-* mice when the gastrointestinal steps of pathogenesis were bypassed and mice were infected i.v. with a much lower dose of *Lm* EGDe ($4 \times 10^{5}$ cfu/mouse). The liver bacterial burden was significantly decreased by approximately 10-fold in *Sdc1-/-* livers compared to Wt livers at 24 h pi (Fig 2D), whereas it was similar at 12 h pi, suggesting that Sdc1 promotes *Lm* pathogenesis after colonization of hepatic tissues. Immunostaining of

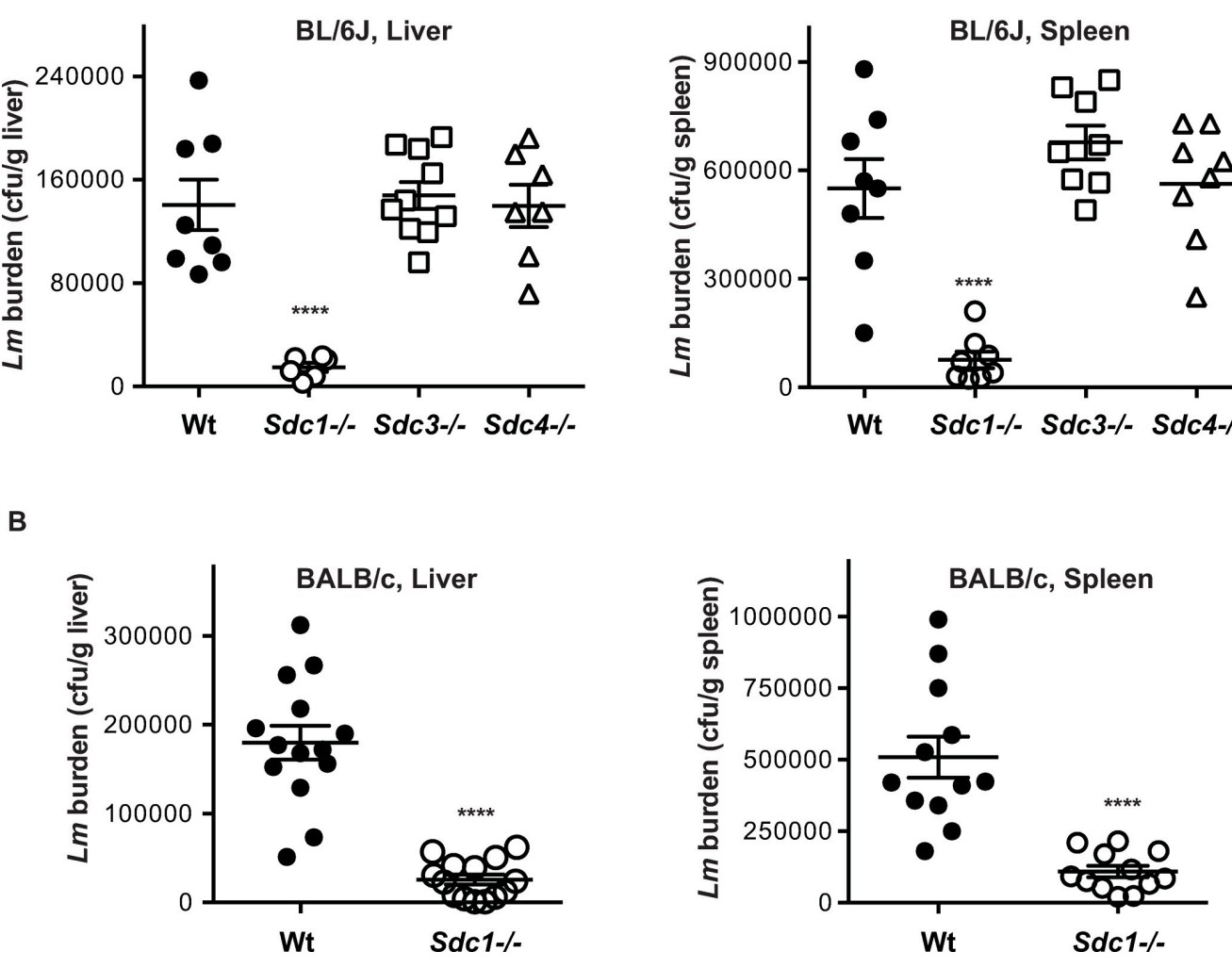

**Fig 1. *Sdc1*-/- mice are significantly less susceptible to i.g. *Lm* infection compared to Wt mice.** A) Wt, *Sdc1*-/-, *Sdc3*-/-, and *Sdc4*-/- mice on the BL/6J background were infected i.g. with $10^{10}$ cfu of *Lm* EGDe and the bacterial burden in the liver and spleen was measured at 24 h pi (n = 6–10, ****$p<0.0001$). B) Wt and *Sdc1*-/- mice on the BALB/c background were infected i.g. with $10^9$ cfu of *Lm* EGDe and the bacterial burden in the liver and spleen was measured at 24 h pi (n = 12–14, ****$p<0.0001$).

infected liver sections for poly <u>N</u>-acetylglucosamine (PNAG), a cell surface polysaccharide expressed by many bacterial pathogens including *Lm* [28], also showed robust accumulation of *Lm* bacteria in Wt livers compared to *Sdc1*-/- livers at 24 h pi (Fig 2E). *Lm* was found dispersed throughout the liver parenchyma and in sinusoidal spaces in Wt livers, whereas substantially fewer *Lm* formed sporadic intravascular aggregates in *Sdc1*-/- liver sinusoids. Together, these results indicate that Sdc1 is not critical for the translocation of *Lm* across the intestinal barrier and the onset of systemic infection. Instead, these results suggest that Sdc1 promotes *Lm* pathogenesis by inhibiting the clearance of *Lm* after it has successfully disseminated to the liver.

## Sdc1 is not a receptor for *Lm*

We next performed adhesion and invasion experiments to directly exclude the role for Sdc1 as a *Lm* receptor. We first examined *Lm* attachment to normal murine mammary gland

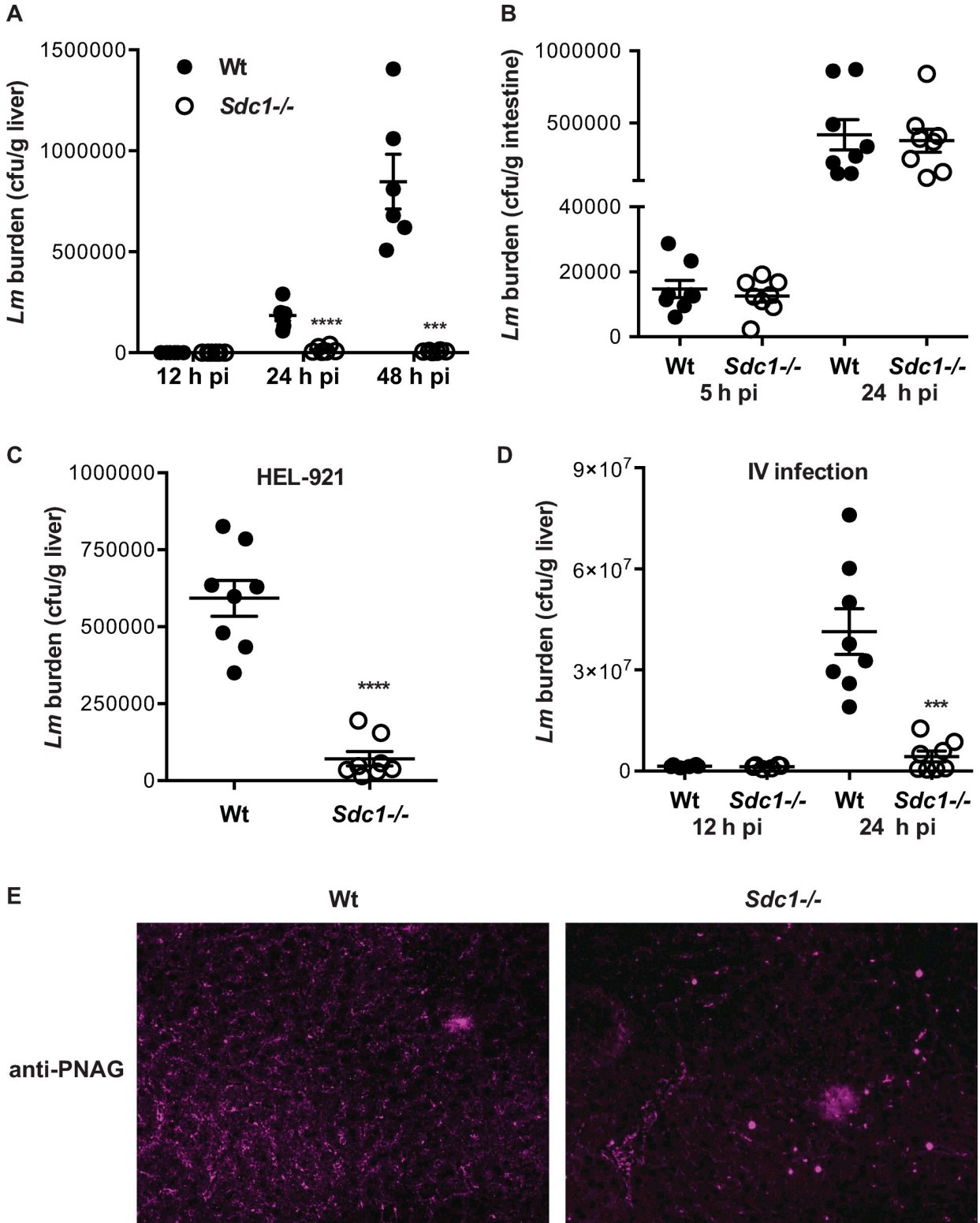

**Fig 2. Sdc1 ablation promotes *Lm* clearance after dissemination into the bloodstream.** A) Wt and *Sdc1-/-* mice on the BALB/c background were infected i.g. with 10^9 cfu of *Lm* EGDe and the liver bacterial burden was measured at 12, 24 and 48 h pi (n = 6, ****p<0.0001 and ***p<0.001). B) Wt and *Sdc1-/-* mice were infected i.g. with 5x10^9 cfu of *Lm* EGDe and the bacterial load in small intestines was measured at 5 and 24 h pi (n = 8). C) Wt and *Sdc1-/-* mice were infected i.g. with 10^9 cfu of *Lm* HEL-921 and the liver bacterial burden was measured at 24 h pi

(n = 8, ****$p < 0.0001$). D) Wt and *Sdc1-/-* mice were infected i.v. with $4 \times 10^5$ cfu of *Lm* EGDe and the liver bacterial burden was measured at 12 and 24 h pi (n = 6–8, ***$p < 0.001$). E) Liver sections of Wt and *Sdc1-/-* mice infected i.v. with $10^5$ cfu of *Lm* EGDe were immunostained with anti-PNAG antibodies (original magnification, x200).

(NMuMG) epithelial cells at 4°C for 1 h. NMuMG cells express the *Lm* receptors E-cadherin [29] and Met [30]. NMuMG cells also express all four syndecans and high levels of Sdc1 (~$10^6$ molecules/cell), and are the prototypical cell type used in various Sdc1 assays [31]. Knockdown of Sdc1 with adenovirus harboring Sdc1 shRNA (Ad-Sdc1 shRNA) decreased cell surface expression of Sdc1 by ~90% in NMuMG cells, but had no inhibitory effect on *Lm* attachment compared to attachment to untreated NMuMG cells or NMuMG cells transduced with control adenovirus (Ad-U6 lamin shRNA) (Fig 3A). Addition of increasing doses of purified Sdc1 ectodomain, HS, CS, or core protein devoid of HS and CS chains to compete out potential Sdc1 interactions also had no inhibitory effect on *Lm* adhesion onto NMuMG cells (Fig 3A).

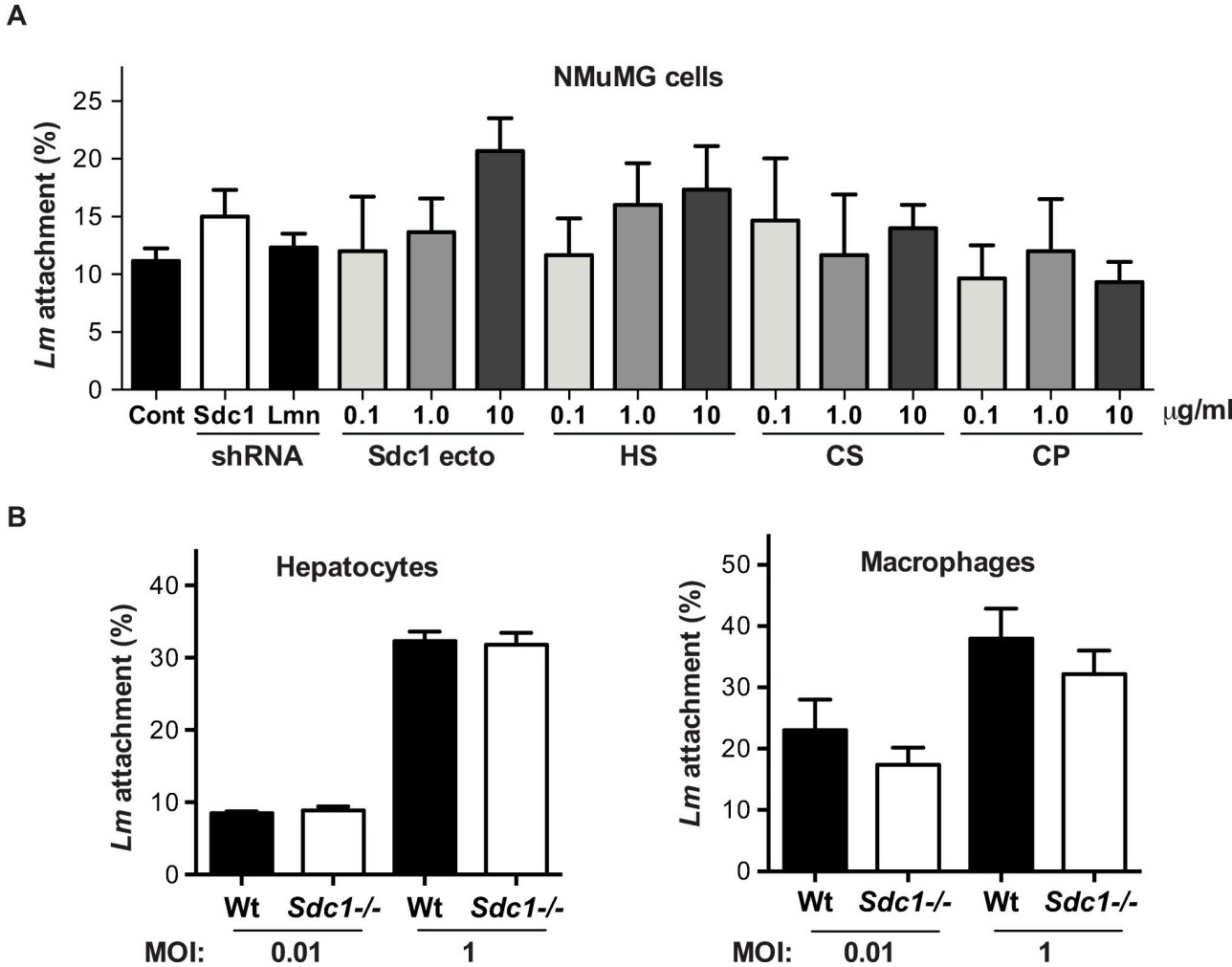

**Fig 3. Sdc1 is not an attachment receptor for *Lm*.** A) Confluent NMuMG cells in 96 well plates pretreated without (control) or with Ad-Sdc1 shRNA or Ad-U6 lamin (Lmn) were incubated with $5 \times 10^3$ cfu of *Lm* EGDe (MOI = 0.1) for 1 h at 4°C in the absence or presence of 0.1 or 10 μg/ml purified Sdc1 ectodomain, HS, CS, or core protein (CP). Attached bacteria were quantified by plating serial dilutions of detergent extracts onto BHI agar plates (n = 3). B) Primary Wt and *Sdc1-/-* hepatocytes and macrophages in 24 well plates were incubated with $2 \times 10^3$ (MOI = 0.01) or $2 \times 10^5$ (MOI = 1) cfu of *Lm* for 1 h at 4°C and attached bacteria were quantified by plating serial dilutions of detergent extracts onto BHI agar plates (n = 5).

We also found a similar number of internalized bacteria when NMuMG cells were incubated with *Lm* without or with excess Sdc1 ectodomain, HS, CS, or core protein for 2 h at 37˚C and non-internalized bacteria were removed by gentamycin incubation (S4A Fig).

We next examined *Lm* attachment and internalization in isolated primary Wt and *Sdc1-/-* cells. While our studies indicated that Sdc1 does not regulate *Lm* infection of intestinal epithelial cells, *Lm* can infect several other host cells, including hepatocytes and macrophages [32]. Unlike InlA that binds to human E-cadherin but not to mouse E-cadherin, InlB can bind to both human and mouse c-Met and C1qR on hepatocytes and macrophages, respectively [33]. Furthermore, all epithelial cells express abundant Sdc1 on their cell surface [3, 24] and Sdc1 expression can be induced on the surface of macrophages [34]. We incubated primary Wt and *Sdc1-/-* hepatocytes and macrophages with *Lm* at MOIs of 0.01 or 1 for 1 h at 4˚C, and quantified the number of attached bacteria. *Lm* showed a dose-dependent increase in binding to primary hepatocytes and macrophages, but the extent of *Lm* adhesion was similar between primary Wt and *Sdc1-/-* cells (Fig 3B). Consistent with these findings, *Lm* invasion of primary Wt and *Sdc1-/-* hepatocytes and macrophages was also similar (S4B Fig). These data indicate that Sdc1 does not promote *Lm* pathogenesis by facilitating bacterial adhesion and invasion of target host cells.

## Sdc1 inhibits neutrophils and complement to promote listeriosis

Subsequent studies were performed with the i.v. model of listeriosis because our data indicated that Sdc1 promotes *Lm* pathogenesis after the onset of systemic infection. We examined if Sdc1 promotes listeriosis by inhibiting host defense mechanisms because our results suggested that Sdc1 interferes with the clearance of bacteria after they have disseminated to hepatic tissues. While the ability to invade host cells and to escape extracellular defense mechanisms is considered as the primary virulence activity of *Lm* that leads to fulminant infection with high mortality, *Lm* is a pathogen that has adapted to intracellular infection and is not an obligate intracellular bacterium. At any given time during *Lm* pathogenesis, a significant number of extracellular bacteria are present in tissues, especially at early phases of pathogenesis. In fact, several arms of innate immunity mediated by inflammatory cytokines, the complement system, monocytes, macrophages, and neutrophils have been shown to slow the progression of infection prior to the development of T cell-mediated immunity [35]. The observation that T cell-deficient mice are able to effectively control the early phase of infection further points to the importance of innate mechanisms in limiting *Lm* infection [36].

To examine if the reduced susceptibility of *Sdc1-/-* mice to listeriosis is attributable to enhanced innate immune responses, we tested the effects of depleting macrophages with clodronate liposomes and complement with cobra venom factor, and immunodepleting neutrophils with anti-Ly6G antibodies and T cells with anti-CD3 antibodies on *Lm* infection in *Sdc1-/-* mice. Depletion of macrophages and CD3+ T cells prior to *Lm* infection increased the liver bacterial burden by 6- and 4-fold at 24 h pi compared to untreated *Sdc1-/-* mice but the difference did not reach statistical significance (Fig 4). However, both neutrophil and complement depletion significantly increased the liver bacterial burden by more than 35-fold (Fig 4). These data indicate that both neutrophils and complement are important in the clearance of *Lm* in *Sdc1-/-* mice. These results also suggest that Sdc1 inhibits neutrophil and complement activities to promote *Lm* pathogenesis.

## Sdc1 does not regulate neutrophil recruitment in listeriosis

Neutrophils do not express Sdc1, and Wt and *Sdc1-/-* neutrophils express similar levels of the differentiation marker Ly6G, and the chemoattractant receptors CXCR2 and C5aR on their

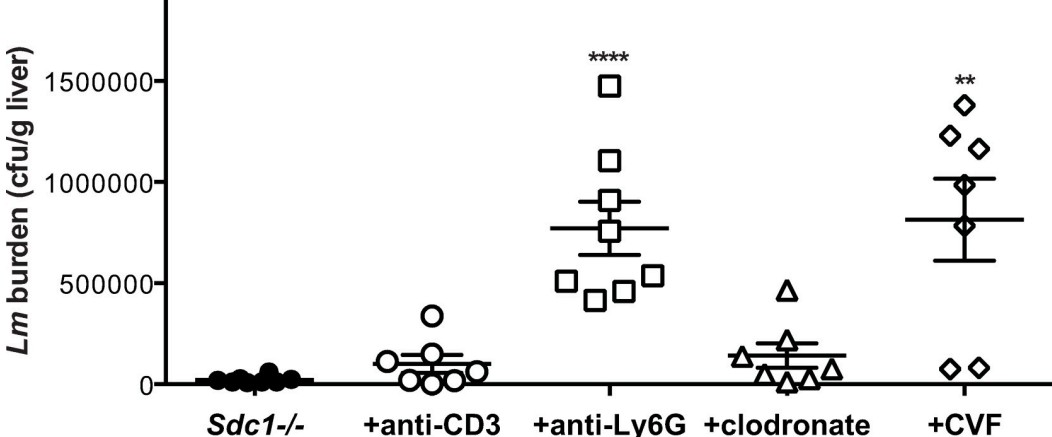

**Fig 4. Complement and neutrophil depletion enhances *Lm* virulence in *Sdc1-/-* mice.** *Sdc1-/-* mice were pre-treated with PBS, 2.5 mg/kg anti-CD3 or -Ly6G antibodies, 50 g/kg clodronate liposomes, or 1 mg/kg CVF and infected i.v. with $2x10^5$ cfu of *Lm* EGDe. The liver bacterial burden was assessed at 24 h pi (n = 7–8, $^{****}p<0.0001$, $^{**}p<0.01$).

cell surface. Wt and *Sdc1-/-* neutrophils also migrate similarly when stimulated with formyl peptides *in vivo* [37] and similarly kill *S. aureus in vitro* [38], indicating that *Sdc1-/-* neutrophils do not have inherent defects in their ability to differentiate, migrate, or kill bacteria. However, Sdc1 can modulate neutrophil inflammation by regulating the expression or activity of neutrophil chemoattractants [37, 39]. We therefore examined if enhanced neutrophil responses mediate the decreased susceptibility of *Sdc1-/-* mice to listeriosis by testing the effects of neutralizing antibodies against CXCR2 or C5aR. *Sdc1-/-* mice were infected i.v. with *Lm* EGDe and immediately injected i.p. with 3.75 mg/kg of blocking anti-CXCR2 or -C5aR antibodies, and the liver bacterial burden and neutrophil accumulation in livers were determined at 24 h pi. Compared to mice infected with *Lm* alone, mice given anti-CXCR2 and anti-C5aR antibodies showed a 5.5-fold and 23-fold increase in liver cfu counts (Fig 5A). Mice treated with anti-CXCR2 and anti-C5aR antibodies also showed a concomitant 2.1-fold and 77-fold decrease in

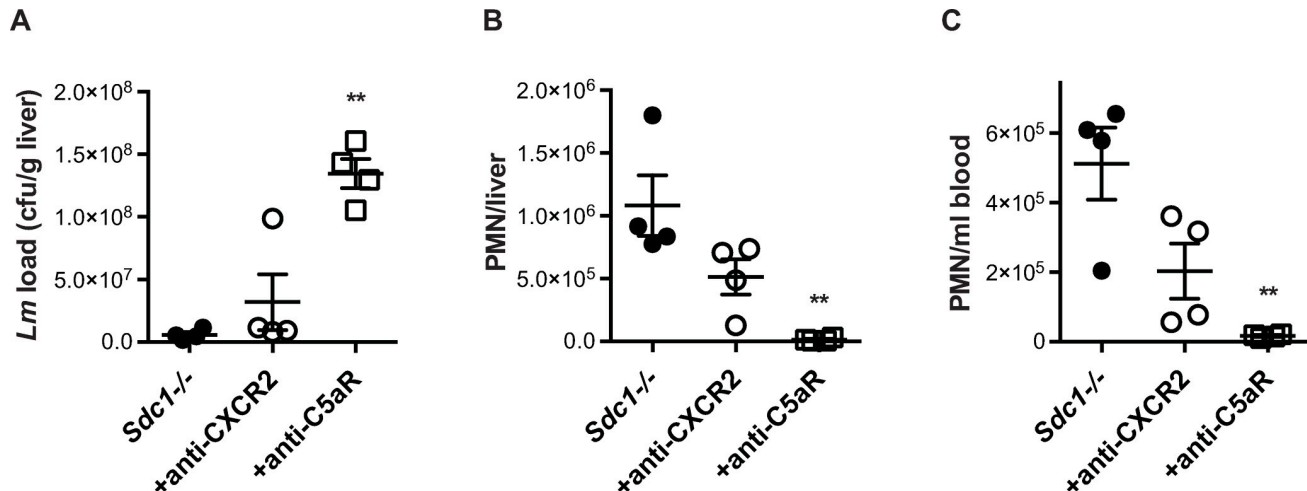

**Fig 5. C5aR inhibition enhances *Lm* virulence in *Sdc1-/-* mice.** *Sdc1-/-* mice were infected i.v. with $7x10^5$ cfu of *Lm* EGDe and immediately injected i.p. with 3.75 mg/kg of neutralizing anti-CXCR2 or anti-C5aR antibodies. A) The liver bacterial burden, B) liver neutrophils, and C) blood neutrophils were determined at 24 h pi (n = 4, $^{**}p<0.01$).

neutrophil accumulation in livers (Fig 5B). However, both anti-CXCR2 and anti-C5aR antibodies also decreased the level of circulating neutrophils. The concentration of circulating neutrophils in mice given anti-CXCR2 and anti-C5aR antibodies was reduced by 2.5-fold and 30.8-fold compared to mice infected with *Lm* alone (Fig 5C). Similar effects were seen when mice were treated with 2.5 mg/kg of the neutralizing antibodies. These observations suggest that anti-CXCR2 and anti-C5aR antibodies promote listeriosis in *Sdc1-/-* mice not by inhibiting neutrophil recruitment to infected livers, but by fixing complement on the neutrophil surface and inducing complement-mediated neutrophil lysis. These data indicate that the dose of migration-blocking antibodies need to be carefully titrated when used *in vivo* to achieve the intended specific inhibitory effect on neutrophil recruitment. Regardless, these data confirm that neutropenia significantly increases the susceptibility of *Sdc1-/-* mice to listeriosis, and further demonstrate that neutrophils are the primary cellular target of Sdc1.

To figure out how Sdc1 inhibits neutrophils to promote listeriosis, we next examined if there are differences in the number of neutrophils recruited to *Wt and Sdc1-/-* livers. When directly counted, the number of accumulated neutrophils increased similarly to approximately $4.3 \times 10^5$ and $5.3 \times 10^5$ in Wt and *Sdc1-/-* livers at 24 h pi, accounting for 77% and 72% of total leukocytes in livers (Fig 6A). FACS analysis also showed that the proportion of neutrophils in Wt and *Sdc1-/-* livers is similar (Fig 6B). Furthermore, the proportion of circulating neutrophils was increased but also similar in Wt and *Sdc1-/-* mice at 24 h pi (Fig 6C), indicating that Sdc1 ablation does not affect the mobilization of neutrophils from the bone marrow or the number of neutrophils recruited to the liver during *Lm* infection. Consistent with these results, levels of neutrophil chemoattractants in blood and liver homogenates were also similar between Wt and *Sdc1-/-* mice. Protein levels of KC (CXCL1), C5a, and IL-17A were similar in Wt and *Sdc1-/-* liver homogenates and serum at 12 and 24 h pi (S5 Fig). In addition, the chemoattractant receptors CXCR2 and C5aR were expressed at similar levels and proportion on the surface of circulating Wt and *Sdc1-/-* neutrophils at 12 h pi (S6 Fig). These data demonstrate that Sdc1 does not affect the expression of neutrophil chemoattractants and their receptors on neutrophils, and the quantity of neutrophils recruited to the liver in listeriosis.

## Sdc1 ablation enhances the formation of intravascular aggregates of neutrophils and NETs

We next immunostained infected Wt and *Sdc1-/-* liver sections with anti-Ly6G antibodies to determine if there are qualitative differences in how neutrophils are recruited. We noticed that neutrophils form large intravascular aggregates in infected *Sdc1-/-* livers, whereas the aggregates were substantially smaller and fewer in infected Wt livers (Fig 7A and 7B). The intravascular neutrophil aggregates resembled the *Lm* clusters seen in infected *Sdc1-/-* livers (Fig 2E), suggesting that neutrophils colocalize with *Lm*. Indeed, coimmunostaining for *Lm* with anti-PNAG antibodies showed that *Lm* colocalizes with neutrophils in the large intravascular lesions of *Sdc1-/-* livers (Fig 7B). Because intravascular neutrophil aggregation is associated with the formation of neutrophil extracellular traps (NETs) in septic mice [40], we also immunostained for histones with anti-pan histone antibodies and found that histones also colocalize with neutrophil-*Lm* clusters (Fig 7B). Moreover, coimmunostaining for other NET components showed robust staining for the cathelicidin antimicrobial peptide CRAMP (S7A Fig), antimicrobial protein MPO (S7B Fig), and also citrullinated histone H4 (S7C Fig) in the intravascular neutrophil-*Lm* aggregates. These results indicate that the large intravascular lesions in *Sdc1-/-* livers are aggregates of neutrophils, *Lm*, and NETs embedded with antimicrobial factors. These data also suggest that the trapping and killing of *Lm* by hepatic intravascular NETs underlie how *Sdc1-/-* mice resist listeriosis, and that Sdc1 inhibits this unique host defense mechanism.

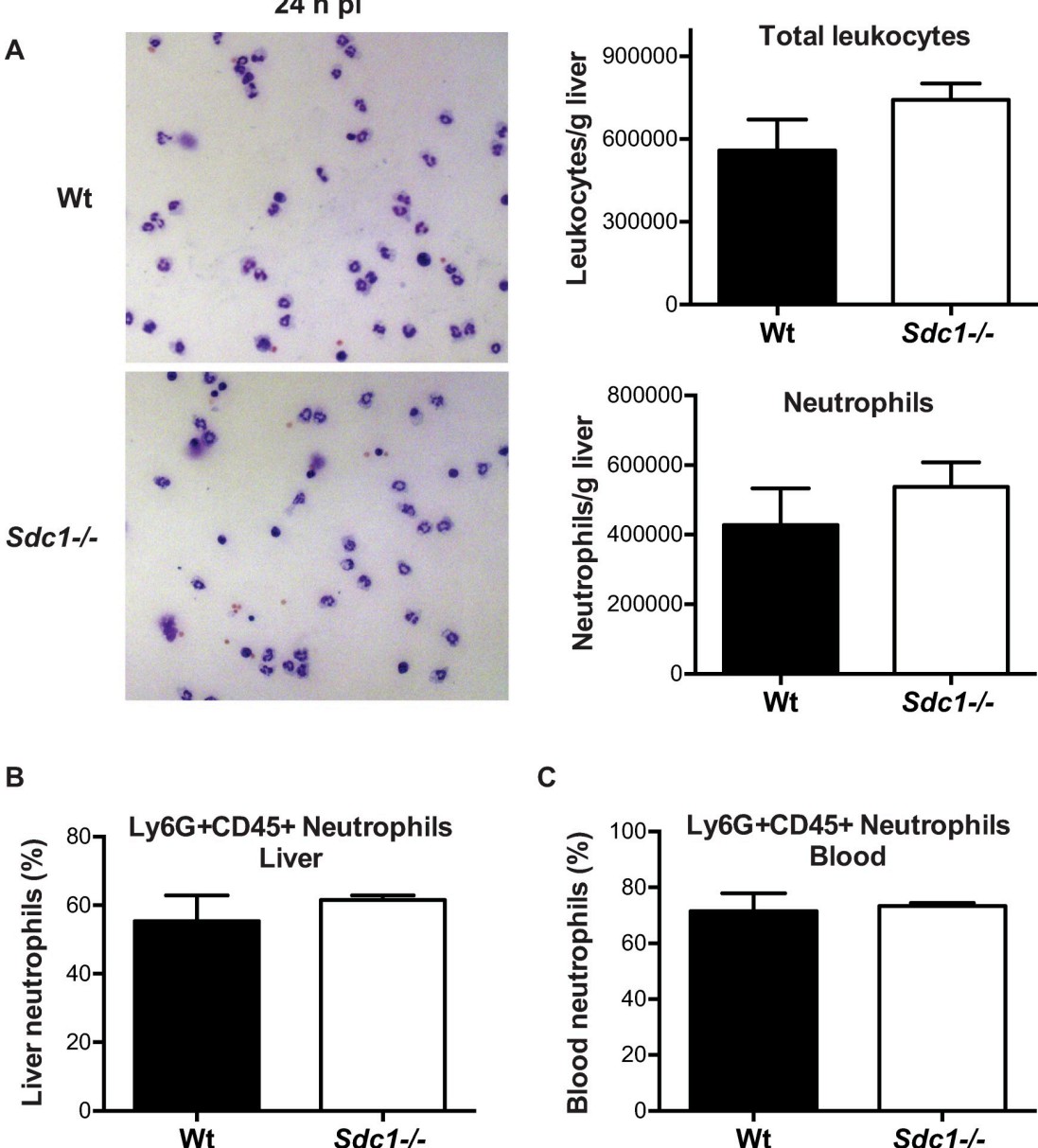

**Fig 6. Sdc1 ablation does not regulate the accumulation of neutrophils in livers of *Lm*-infected mice.** A) Wt and *Sdc1-/-* mice were infected with 5x10$^5$ cfu of *Lm* EGDe and neutrophil accumulation in livers was measured at 24 h pi by staining liver leukocyte fractions spun down on cytospin slides with HEMA-3 stain and directly counting total leukocytes and neutrophils (n = 3). Representative HEMA-3 stained Wt and *Sdc1-/-* liver leukocytes are shown. B) Proportion of Ly6G+CD45+ neutrophils in total liver leukocytes of Wt and *Sdc1-/-* mice was determined at 24 h pi by FACS (n = 3). C) Proportion of circulating Ly6G +CD45+ neutrophils in Wt and *Sdc1-/-* mice was determined at 24 h pi by FACS (n = 3).

## *Lm* infection induces Sdc1 shedding and Sdc1 ectodomains inhibit intravascular NETs to promote listeriosis

We next examined if *Lm* exploits Sdc1 shedding to enhance its virulence because our data indicated that cell surface Sdc1 is not a *Lm* receptor and previous studies have shown that Sdc1 ectodomains can function as soluble inhibitors of neutrophilic inflammatory responses [38, 39]. *Lm* infection significantly and specifically stimulated Sdc1 shedding by increasing serum

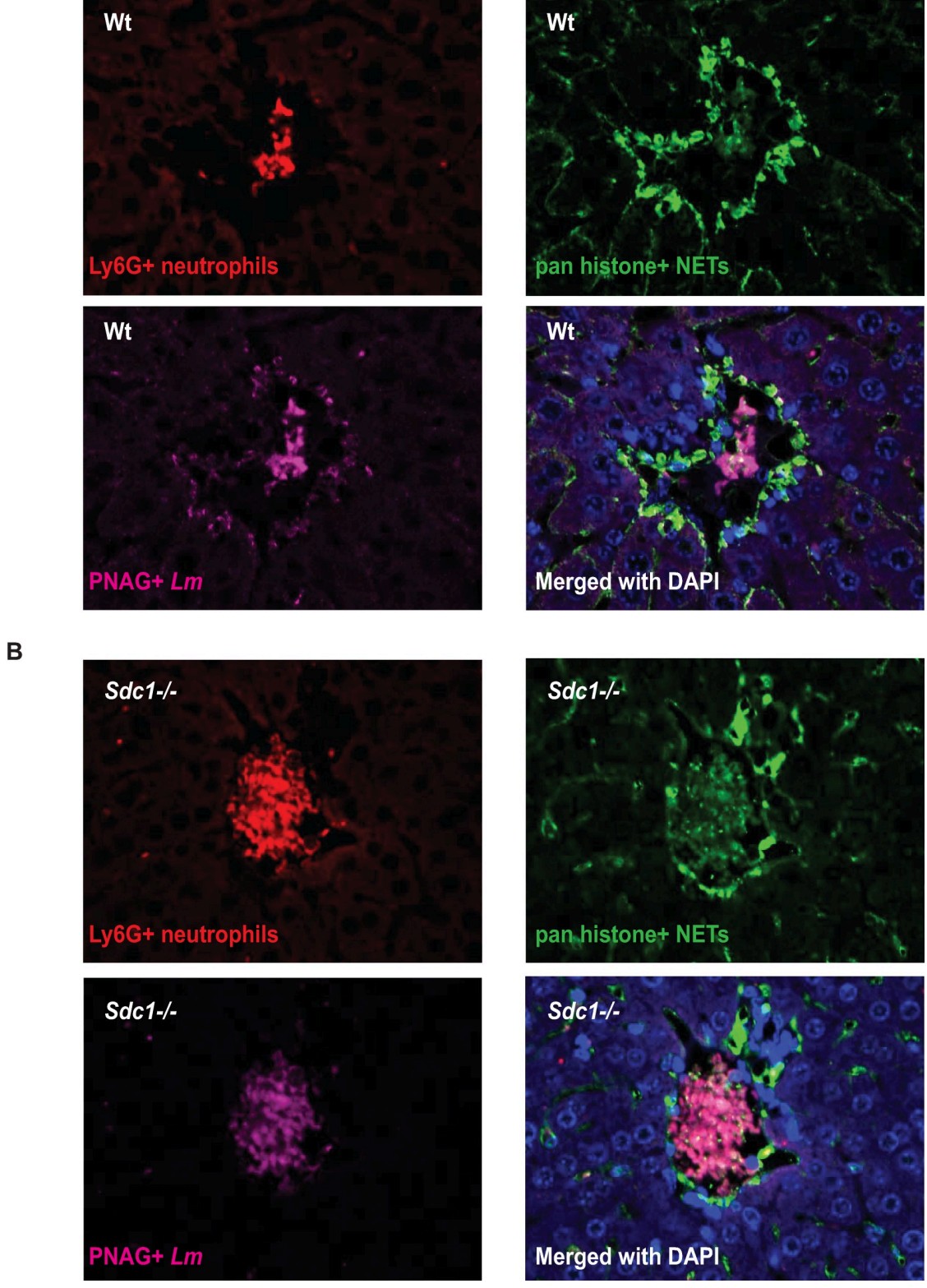

**Fig 7. Sdc1 ablation enhances the formation of intravascular aggregated neutrophils and NETs in listeriosis.** A) Wt and B) *Sdc1-/-* liver sections (24 h pi) were immunostained for neutrophils (anti-Ly6G), NETs (anti-pan histones), and *Lm* (anti-PNAG). Merged pictures show counterstaining with DAPI (original magnification, x200).

levels of Sdc1, but not Sdc4 ectodomains, in Wt mice (S8A Fig). Serum Sdc1 ectodomains reached a maximum at 15-fold over baseline at 6 h pi and high levels were sustained at 24 h pi. Serum Sdc4 ectodomains were barely above baseline at all times examined. Furthermore, levels of cell-associated Sdc1 in total liver extracts of infected Wt mice were decreased by 4-fold at 6 h pi, whereas Sdc1 levels were largely unaffected in the spleen and lung (S8B Fig). In addition, Sdc1 expression on the sinusoidal surface of hepatocytes was markedly reduced in infected Wt livers (S8C Fig). These data indicate that Sdc1 ectodomains are primarily shed from hepatocytes during listeriosis.

We next examined if the absence of Sdc1 ectodomains underlies the decreased susceptibility of *Sdc1-/- mice* to listeriosis by injecting purified Sdc1 ectodomains into *Sdc1-/-* mice at 6 h pi, a time point where *Lm* infection-induced Sdc1 shedding is maximal in Wt mice. Delayed i.v. administration of 2 µg of purified Sdc1 ectodomains or HS, but not CS or core protein, significantly increased the liver bacterial burden in *Sdc1-/-* mice by 31-fold and 27-fold at 24 h pi (Fig 8A), indicating that Sdc1 ectodomains promote *Lm* infection in an HS-dependent manner. To pursue the essential modification of HS that enhances *Lm* virulence, we tested the effects of chemically desulfated heparin. Administration of unmodified and desulfated heparin compounds showed that 2-*O*-desulfation significantly reduces heparin's ability to increase the liver bacterial burden (Fig 8B). *N*-desulfation also partially inhibited heparin's capacity to increase the bacterial load, but 6-*O*-desulfation did not (Fig 8B). These data indicate that *Lm* induces Sdc1 shedding to exploit the activity of 2-*O*-sulfated, and possibly also *N*-sulfated HS motifs in Sdc1 ectodomains to promote its pathogenesis.

We next tested if Sdc1 ectodomains promote *Lm* pathogenesis by inhibiting intravascular aggregated NETs. Liver sections of *Sdc1-/-* mice infected with *Lm* only or infected with *Lm* and injected with purified Sdc1 ectodomains at 6 h pi were immunostained for neutrophils, histones, and *Lm* at 24 h pi. *Sdc1-/-* livers infected with *Lm* showed sparse but large intravascular aggregates of neutrophils that co-immunostained for histones and *Lm* (Fig 8C). On the other hand, delayed administration of Sdc1 ectodomains substantially reduced the size of aggregated neutrophil and NETs and led to dispersion of *Lm* throughout the liver (Fig 8C), consistent with the increased liver bacterial burden upon Sdc1 ectodomain injection. Similarly, delayed administration of DNase I to degrade the DNA backbone of NETs markedly reduced the size of intravascular neutrophil-histone-*Lm* aggregates and led to widespread dispersion of *Lm* in the liver at 24 h pi (Fig 8C). Delayed injection of DNase I also significantly increased the liver bacterial burden at 24 h pi (S8D Fig), indicating that intravascular aggregated NETs are critical in killing *Lm*. Altogether these results indicate that *Lm* induces Sdc1 shedding to exploit the ability of Sdc1 ectodomains to inhibit intravascular NET-mediated host defense to promote its infection.

## Discussion

We demonstrate here that Sdc1 is a critical host factor that promotes listeriosis. However, despite many evidence supporting the role of cell surface HSPGs in microbial attachment and invasion, our studies indicate that Sdc1 does not enhance *Lm* virulence in this manner. Instead, Sdc1 is shed from the cell surface upon *Lm* infection. While the pathogenic reason for *Lm* to activate Sdc1 shedding to inhibit host defense is apparent, the compelling reason for not using Sdc1 as an adhesion and invasion receptor is not obvious, given the fact that Sdc1 is a predominant cell surface HSPG on host cells targeted by *Lm*. Perhaps the abundance of Sdc1 on the surface of target cells impedes, rather than promotes, *Lm* attachment and invasion. For example, because Sdc1 outnumbers Met in hepatocytes, Sdc1 binding may scavenge *Lm*, interfere with *Lm* InlB binding to Met, and inhibit subsequent Met signaling required for bacterial

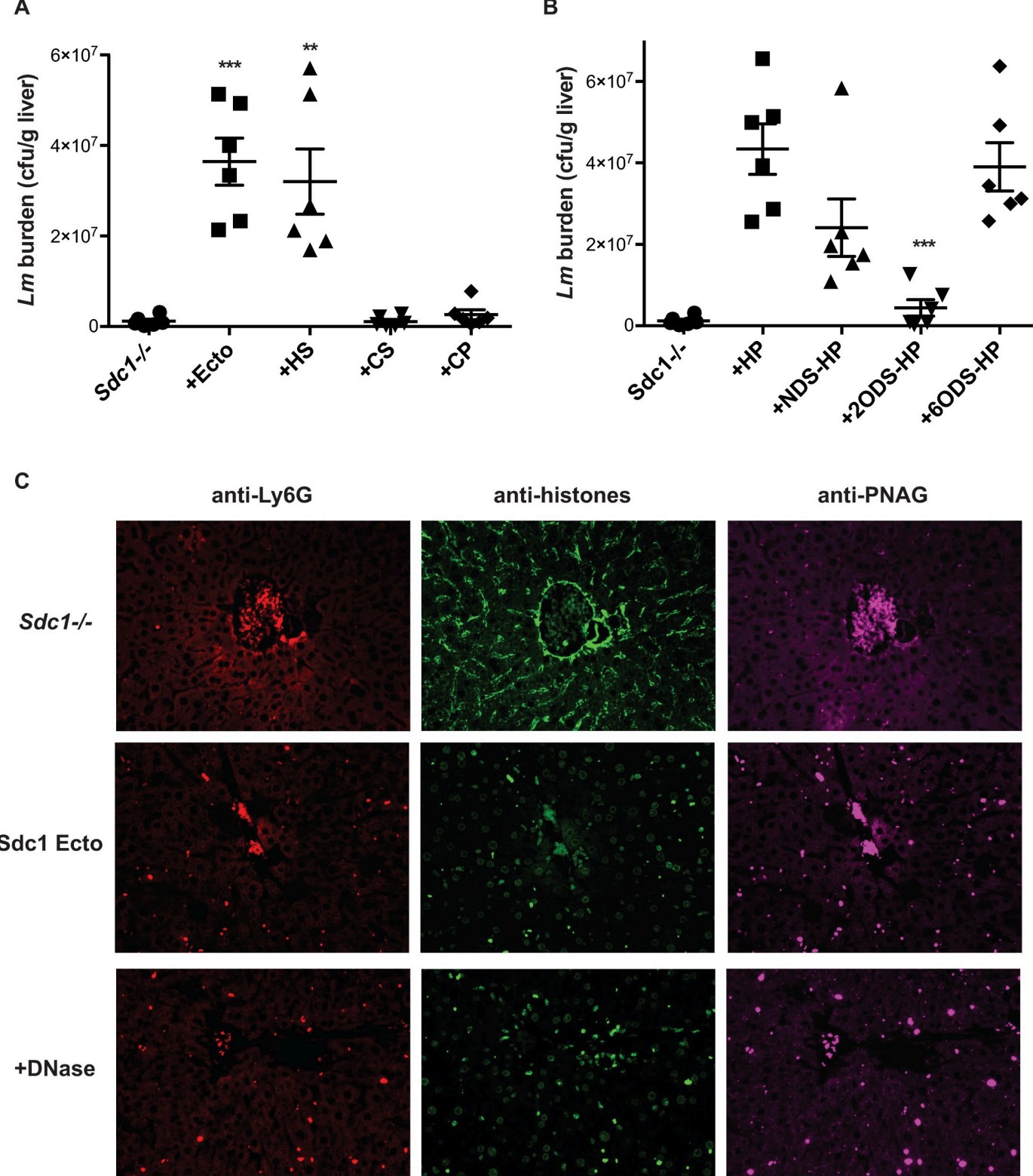

**Fig 8. Sdc1 ectodomains promote listeriosis by inhibiting intravascular aggregated NETs.** *Sdc1-/-* mice were infected i.v. with 2x10⁵ cfu of *Lm* and injected without or with A) 2 μg/mouse purified Sdc1 ectodomains (Ecto), HS, CS, or core protein devoid of HS and CS (CP) or B) 2 μg/mouse heparin (HP), *N*-desulfated HP (NDS-HP), 2-*O*-desulfated HP (2ODS-HP), or 6-*O*-desulfated HP (6ODS-HP) at 6 h pi and the liver bacterial burden was determined at 24 h pi (n = 6, ***p<0.001, **p<0.01). C) *Sdc1-/-* mice were infected i.v. with 4.5x10⁵ cfu of *Lm* and injected without or with 2 μg/mouse purified Sdc1 ectodomains at 6 h pi or 500 U/mouse

DNase I at 12 h pi and livers were isolated at 24 h pi. Liver sections were immunostained with anti-Ly6G, -pan histones, and -PNAG antibodies (original magnification, x200).

internalization. In fact, stimulation of Sdc1 shedding by heparanase enhances Met signaling in myeloma cells [41]. Furthermore, not all HSPGs are created equal. HS chains of HSPGs are attached to specific locations in core proteins and also differ in terms of size, charge density and degree of sulfation, suggesting that elaboration of HS motifs recognized by *Lm* may be restricted to a specific subset of HSPGs. While structural features of HSPGs that potentiate InlB-Met signaling have not been studied in detail, HSPGs such as CD44 and glypican-4 can enhance Met activation by HGF, but not glypican-1 or glypican-3 [42, 43], suggesting that similar differences in the ability to potentiate Met signaling by InlB may exist among HSPGs. Together, these observations suggest that high expression of Sdc1 may interfere with *Lm* interactions with host cells and that removal of Sdc1 from the cell surface by ectodomain shedding may facilitate *Lm* interactions with other cell surface HSPGs. Additional studies with specific inhibitors of *Lm*-induced Sdc1 shedding or cells and mice expressing an uncleavable form of Sdc1 are needed to directly determine whether cell surface Sdc1 is indeed detrimental for *Lm* adhesion and internalization.

Our study indicates that stimulation of Sdc1 shedding is an important virulence activity that allows *Lm* to evade eradication by innate host defense. However, how *Lm* induces Sdc1 shedding remains to be defined. *Lm* infection specifically activates Sdc1 shedding from the surface of hepatocytes, suggesting that direct *Lm* interactions with hepatocytes are required. In this regard, Met signaling by InlB may be important because activation of receptor tyrosine kinases have been shown to enhance Sdc1 shedding [44]. Alternatively, because several pore-forming bacterial toxins stimulate Sdc1 shedding by epithelial cells [45, 46], cytolytic toxins of *Lm* such as listeriolysin O (LLO), may function as a Sdc1 shedding enhancer. Anthrax ALO, a similar cholesterol-dependent cytolysin, has been shown to activate Sdc1 shedding [46] and LLO can be secreted extracellularly [47]. However, since the optimal pH for LLO's cytolytic activity is 5.5, it is uncertain if the toxin is capable of stimulating Sdc1 shedding in the neutral pH range of extracellular compartments.

Upon infection with *Lm*, a multifaceted innate immune response is initiated. Although the contribution of neutrophils to listeriosis has been investigated for decades, their importance and how they defend against *Lm* infections are only just becoming understood. Early studies with anti-GR1 antibody to immunodeplete neutrophils could not establish the role of neutrophils in listeriosis because anti-GR1 reacts with both Ly6G expressed on neutrophils and Ly6C on monocytes, subsets of dendritic cells, macrophages and T cells [48–50]. However, subsequent immunodepletion studies using anti-Ly6G antibodies prior to *Lm* challenge clearly showed that neutrophils are required to eradicate *Lm* at early stages of listeriosis, especially in the liver [51, 52]. The importance of neutrophils in listeriosis is also supported by studies showing that mutations that induce neutropenia or impair neutrophil functions increase the susceptibility of mice to listeriosis. For example, mice lacking G-CSF or its receptor become neutropenic and are hypersusceptible to *Lm* infection [53, 54]. Complement C5-deficient A/J mice are hypersusceptible to listeriosis because of severely dampened neutrophil and macrophage responses [55]. Furthermore, cancer patients presenting neutropenia are more susceptible to listeriosis [56]. On the other hand, mice lacking the IFNAR receptor for type I interferons [57] or the integrin LFA-1 [58] are less susceptible to listeriosis because of an increased infiltration of neutrophils into the site of infection.

In agreement with these observations, our studies indicate that neutrophils are required for *Sdc1-/-* mice to significantly resist *Lm* infection. However, Sdc1 does not affect the rapid and

robust mobilization of neutrophils to *Lm*-infected livers. Instead, our studies provide both genetic and biochemical evidence that Sdc1 inhibits the clearance of *Lm* by intravascular aggregated NETs. In most tissues, neutrophils are generally recruited through the post-capillary venules. An exception is the liver where an overwhelming majority of neutrophils can be recruited via the capillary-like sinusoids [59], suggesting that neutrophils recruited to the liver are well positioned to mount an intravascular host response to combat systemic infections such as listeriosis. Furthermore, NETs can be formed under slow-flowing conditions that simulate those of the liver sinusoids and the quantity of bacteria that can be captured by NETs is considered to far exceed that can be captured by individual neutrophils [60].

However, what comes first and the chronology of how intravascular aggregated NETs are formed *in vivo* in response to *Lm* infection remains to be determined. NET formation can be induced by pathogenic bacteria that can survive and escape phagosomal killing [61]. The size of microbes is one of the critical factors for neutrophils in deciding whether to kill pathogens through phagocytic mechanisms or NETs. Larger microbes are more effective at inducing NETs [62], suggesting that NETs may be deployed when the pathogen becomes too large to be phagocytosed. For example, aggregates of *Mycobacterium bovis* drive NET formation in a size-dependent manner [62]. Similarly, *S. aureus* forms aggregates upon exposure to plasma and stimulate NET formation in mouse models of sepsis [63]. These observations suggest that aggregation of *Lm* may be one of the triggers of intravascular aggregated NET formation. Indeed, our preliminary studies suggest that mouse serum can induce the agglutination of *Lm* in a HS-sensitive manner (S9 Fig).

Our studies also suggest that Sdc1 may promote *Lm* pathogenesis by inhibiting complement. *Lm* can dramatically activate complement *in vivo* [64], but direct killing of *Lm* by complement is unlikely because the thick cell wall of Gram-positive *Lm* prevents efficient formation of the complement membrane attack complex [65]. However, complement activation may mediate the formation of intravascular aggregated NETs. One of the major mechanisms of how pathogens activate both lytic and non-lytic NET formation is through activation of complement receptors (CRs) of the β2 integrin family [66]. For example, hantavirus activates NET formation through CR3 ($\alpha_M\beta_2$) and CR4 ($\alpha_X\beta_2$) signaling [67], *Candida albicans* β-glucan induces neutrophil aggregation and NET formation via CR3 in a fibronectin-dependent manner [68], and LPS stimulates NETosis through CR3 [69]. Furthermore, neutrophils aggregate in microvessels under conditions of complement activation [70, 71]. These findings suggest that activation of complement by agglutinated *Lm* leads to neutrophil activation, aggregation, and the formation of aggregated NETs via CRs.

Sdc1 can potentially interfere with several steps of the host defense mediated by intravascular aggregated NETs. First, HS chains of Sdc1 ectodomains may inhibit *Lm* agglutination in the circulation since HS inhibits serum-induced agglutination of *Lm* (S9 Fig). Second, both CR3 and CR4 are heparin-sensitive complement receptors [72, 73], suggesting that Sdc1 ectodomains can also inhibit CR3 and CR4 interactions required for NET formation. However, Sdc1 ectodomains are unlikely to directly inhibit intracellular processes that lead to NET formation because highly anionic ectodomains are membrane-impermeable. Third, because the pI of HS is approximately 2–3, whereas that of DNA is about 5, HS chains of Sdc1 ectodomains are likely to interfere with the interaction between DNA and cationic antimicrobial compounds in NETs, and possibly displace the antimicrobials from NET fibers. Furthermore, Sdc1 binding inhibits bacterial killing by cationic antimicrobials [74, 75]. These functions of Sdc1 in regulating NETs are by no means exclusive. Considering the versatile ability of Sdc1 HS to bind specifically to a large number of biological molecules, Sdc1 ectodomains are likely to possess several mechanisms to inhibit intravascular aggregated NETs. While the clinical importance of NETs in the host defense against *Lm* remains to be established, neutrophils of individuals who are more susceptible to listeriosis, such as neonates and elderly individuals,

have an impaired capacity to form NETs [76–79]. These findings suggest that the ability of *Lm* to stimulate Sdc1 shedding and subvert Sdc1 ectodomains to inhibit intravascular aggregated NETs may tip the delicate balance between host defense and pathogenesis in listeriosis.

## Materials and methods

### Ethics statement

All animal experiments were approved by the Institutional Biosafety Committee and Institutional Animal Care and Use Committee of Boston Children's Hospital, and complied with federal guidelines for research with experimental animals.

### Materials

Tryptic soy agar (TSA), brain heart infusion (BHI) agar, and BHI broth were from BD Biosciences (San Jose, CA). Porcine mucosa HS, heparin, and chemically desulfated heparin compounds were purchased from Neoparin (Alameda, CA). Bovine tracheal CS-A was purchased from Sigma-Aldrich (St. Louis, Missouri, USA). Native Sdc1 ectodomains were purified from the conditioned medium of normal murine mammary gland (NMuMG) epithelial cells by DEAE chromatography, CsCl density centrifugation, and immunoaffinity chromatography as described previously [38, 80]. Recombinant mouse Sdc1 core protein devoid of HS and CS were expressed as a GST fusion protein in *E. coli* and purified by glutathione affinity chromatography as described previously [75]. Immobilon Ny+ (cationic nylon membrane) was from Millipore (Bedford, MA). DNase I was purchased from Worthington Biochemical (Lakewood, NJ). ViraPower Adenovirus Expression System, AccuPrime Pfx DNA polymerase, pENTR/ U6, pAd/Block IT-DEST, LR clonase II, Lipofectamine 2000, and 293A cells were from Invitrogen (Carlsbad, CA). FAST-TRAP virus purification and concentration kit was from EMD Millipore (Billerica, MA) and Adeno-X Rapid Titer kit was from Clontech (Mountain view, CA). Oligonucleotide primers were from IDT (Coralville, IA). ELISA kits for mouse KC, C5a, and IL-17A were from R&D Systems (Minneapolis, MN). Clodronate liposomes (50 g/kg i.v. for macrophage depletion) were from Encapsula Nanosciences (Brentwood, NJ). Cobra venom factor (CVF, 1 mg/kg i.p. for complement depletion) was purchased from Quidel (San Diego, CA). All other materials except for the immunochemicals described below were purchased from Thermo Fisher Scientific (Waltham, MA), Sigma, or VWR (Westchester, PA).

### Immunochemicals

Rat anti-mouse Sdc1 ectodomain (281.2, 5 μg/ml for IHC, 1 μg/ml for immunoblotting), rat anti-mouse Ly6G (1A8, 2.5 mg/kg i.p. for immunodepletion), and rat-anti-mouse C5aR (20/ 70, 2.5 or 3.75 mg/kg i.p. for neutralization) monoclonal antibodies were purchased from Biolegend (San Diego, CA). Rat anti-mouse CD3 (17A2, 2.5 mg/kg i.p. for immunodepletion) and rat anti-mouse Sdc4 ectodomain monoclonal antibodies (Ky8.2, 1 μg/ml for immunoblotting) were from Bio X cell (West Lebanon, NH) and BD Biosciences, respectively. Rat anti-mouse CXCR2 (242216, 2.5 or 3.75 mg/kg i.p. for neutralization) antibody was from R&D Systems (Minneapolis, MN) and affinity-purified rabbit anti-citrullinated histone H4 antibody (07– 596, 1:200 for IHC) was from Sigma. Protein A/G-purified abbit anti-mouse Sdc2 (MSE-2, 1 μg/ml for immunoblotting) and anti-mouse Sdc3 (MSE-3, 1 μg/ml for immunoblotting) polyclonal antibodies were generated as described previously [81]. Pan anti-histone antibody was from US Biologicals (Salem, MA). Rabbit anti-MPO polyclonal antibody (L607, 1:100 for IHC) was from Cell Signaling (Danvers, MA). Affinity-purified rabbit anti-mature CRAMP (10 μg/ml for IHC) was generated in-house. Human anti-PNAG monoclonal antibody (F598,

1:1000 for IHC) was a generous gift from Dr. Gerald Pier (Harvard Medical School, Boston, MA). Horseradish peroxidase-conjugated and Alexa Fluor-conjugated secondary antibodies were purchased from Jackson ImmunoResearch (West Grove, PA) or Biolegend and used at recommended dilutions.

## Bacteria

*Lm* strain EGDe was from ATCC (BAA-679) and *Lm* strains 10403S and HEL-921 were kindly provided by Dr. Darren Higgins (Harvard Medical School, Boston, MA). EGDe was grown to mid-log phase in brain heart infusion (BHI) broth at 37°C with agitation (250 rpm), whereas 10403S and HEL-921 were grown in BHI broth at 30°C without agitation. The bacterial concentration was estimated by turbidity (OD600 nm). Bacteria were washed and resuspended in PBS for infection assays *in vivo* and in culture medium for cell-based attachment and internalization assays. The viable infectious inoculum was determined by plating out serial dilutions onto BHI agar plates.

## Mice

Unchallenged *Sdc1-/-* mice on the BALB/c and C57BL/6J backgrounds, and *Sdc3-/-* and *Sdc4-/-* mice on the C57BL/6J background are healthy with normal growth, reproduction, tissue morphology, and CBC and serum chemistry parameters. Both female and male *Sdc1-/-*, *Sdc3-/-*, and *Sdc4-/-* mice and corresponding Wt littermates on the BALB/c or C57BL/6J background were used at an age of 6–10 wks. Mice were maintained in microisolator cages under specific pathogen-free conditions in a 12 h light/dark cycle and fed a basal rodent chow *ad libitum*.

## Mouse models of *Lm* infection

Mice were infected with *Lm* using established protocols [82]. Briefly, mice were infected i.g. by oral gavage or i.v. via the tail vein with various doses of *Lm* without or with test reagents in 100 μl PBS. For i.g. infections, mice were fasted for 6 h with free access to water to minimize aspiration of the inoculum during oral infection. The extent of infection was assessed at different times after infection by isolating and weighing tissues, straining tissues through autoclaved stainless steel meshes, and incubating for 30 min at room temperature in BHI broth containing 0.1% Triton X-100. Serial dilutions of tissue homogenates were plated on BHI agar plates to quantify the bacterial burden. To determine the bacterial burden in small intestines, isolated tissues were treated with gentamicin (100 μg/ml for 2 h) prior to homogenization to kill extracellular bacteria and processed for cfu counts. Tissues from non-infected mice were used as baseline controls. In several experiments, mice were pre-treated with anti-Ly6G or -CD3 antibodies, clodronate liposomes, or CVF to deplete specific leukocytes or complement prior to infection, co-infected with blocking anti-CXCR2 or -C5aR antibodies, or post-treated with purified Sdc1 ectodomains, HS, CS, core protein, heparin, desulfated heparin compounds, or DNase I as indicated. To directly quantify leukocytes and neutrophils in livers, isolated livers were homogenized and leukocyte fractions were separated by Percoll density centrifugation. Leukocyte preparations were spun onto cytospin slides and stained with HEMA-3 leukocyte differential stain. Total leukocytes and neutrophils were identified by their morphology and staining pattern, and counted.

## *Lm* attachment and internalization

NMuMG cells and primary Wt and *Sdc1-/-* hepatocytes and macrophages were used to determine the role of Sdc1 in *Lm* attachment and invasion. Adenovirus harboring shRNA against mouse Sdc1 (Ad-Sdc1 shRNA) were generated using the ViraPower Adenovirus Expression

System and NMuMG cells were transduced with the adenoviral vectors as previously described [38]. Primary hepatocytes were isolated from perfused livers by collagenase digestion [83] and primary macrophages were isolated from peritoneal lavages. Confluent NMuMG cells pretreated without or with Ad-Sdc1 shRNA or Ad-U6 lamin shRNA and primary Wt and *Sdc1-/-* hepatocytes and macrophages were incubated with *Lm* at the indicated MOI for 1 hr at 4°C for *Lm* attachment and for 2 h at 37°C for *Lm* internalization measurements in the absence or presence of purified Sdc1 ectodomains, HS, CS, or Sdc1 core protein. In internalization assays, cells were treated with 100 μg/ml gentamycin for 30 min after the incubation period to kill extracellular bacteria. To quantify *Lm* attachment and invasion, host cells were washed and lysed in BHI with 0.1% Triton X-100, and serial dilutions of lysates were plated onto BHI agar plates.

## Syndecan levels in tissues

Tissues isolated from perfused mice were homogenized and washed twice in ice-cold PBS, centrifuged at 5000xg for 20 min, and tissue pellets were resuspended and incubated overnight with 7 M urea with 0.1 M NaCl, protease inhibitors, 0.05 M sodium acetate, pH 6.0 at 4°C to extract proteoglycans. Urea extracts were centrifuged at 16,000xg for 20 min, and proteoglycans were partially purified by DEAE chromatography, dialyzed, and dot blotted onto Immobilon Ny+ membranes. Syndecans were detected with 281.2 anti-Sdc1, MSE-2 anti-Sdc2, MSE-3 anti-Sdc3, or Ky8.2 anti-Sdc4 antibodies and quantified using purified native Sdc1 and Sdc4 ectodomains and recombinant Sdc2 and Sdc3 ectodomains as standards.

## Histopathology

Liver lobes isolated from Wt and *Sdc1-/-* mice before and after *Lm* infection were fixed in 4% paraformaldehyde/PBS for 48 h at 4°C, embedded in paraffin, and sectioned. Liver sections (5 μm) were immunostained with anti-Sdc1, -Ly6G, -histones, -citrullinated histone H4, -CRAMP, -MPO, or -PNAG antibodies as indicated. Images were captured with the Zeiss Axiovert 40 CFL microscope, and pictures were taken with the AxioCam MRm high-resolution camera. Adobe Photoshop CC 2019 was used to process the acquired images.

## Data analyses

Data are expressed as scatterplots with mean±SEM or bar graphs with mean±SEM. Statistical significance between experimental and control groups was analyzed by two-tailed unpaired Student's *t*-test and between multiple groups by ANOVA followed by Dunnett's post-hoc test using GraphPad Prism software (version 6.0e). *P* values ≤0.05 were determined to be significant.

## Supporting information

**S1 Fig. Liver and spleen were isolated from perfused Wt mice, homogenized, and extracted with 7 M urea as described.** Tissue levels of the 4 syndecans in urea extracts were measured by dot immunoblotting (n = 3).
(TIF)

**S2 Fig. Wt and *Sdc1-/-* mice on the BALB/c background were infected i.g. with $10^9$ cfu of *Lm* EGDe and the spleen bacterial burden was measured at 12, 24 and 48 h pi (n = 6, \*\*\****p*<0.001 and \*\****p*<0.01).**
(TIF)

**S3 Fig. Wt and *Sdc1*-/- mice on the BALB/c background were infected i.g. with $10^9$ cfu of *Lm* 10403S and the liver bacterial burden was measured at 24 h pi (n = 5, ***$p<0.001$).** (TIF)

**S4 Fig.** A) Confluent NMuMG cells in 96 well plates were incubated with $4x10^3$ cfu of *Lm* EGDe (MOI = 0.1) for 2 h at 37˚C in the absence or presence of 10 μg/ml purified Sdc1 ectodomain, HS, CS, or core protein (CP). Internalized bacteria were quantified by incubating with 100 μg/ml gentamycin for 30 min to kill extracellular bacteria, washing with PBS, lysing in BHI containing 0.1% Triton X-100, plating out serial dilutions onto BHI agar plates, and counting *Lm* colonies (n = 3). B) Primary Wt and *Sdc1*-/- hepatocytes and macrophages in 96 well plates were incubated with $3x10^3$ *Lm* for 2 h at 37˚C and internalized bacteria were quantified (n = 5).
(TIF)

**S5 Fig. Wt and *Sdc1*-/- mice were infected i.v. with $5x10^5$ cfu of *Lm* EGDe and blood and liver lobes were isolated at 12 and 24 h pi.** Levels of KC, C5a, and IL-17A in liver homogenates and serum were measured by ELISA (n = 4).
(TIF)

**S6 Fig. Wt and *Sdc1*-/- mice were infected i.v. with $7x10^5$ cfu of *Lm* EGDe and expression of CXCR2 and C5aR on circulating neutrophils were measured at 12 h pi by FACS.** MFI plots (n = 3) and representative dot plots of A) CXCR2 and B) C5aR expression on Ly6G+ neutrophils are shown.
(TIF)

**S7 Fig.** Intravascular inflammatory lesions in *Sdc1*-/- liver sections (24 h pi) were immunostained for A) neutrophils (anti-Ly6G), CRAMP, and *Lm* (anti-PNAG), B) neutrophils, MPO, and *Lm*, or C) neutrophils, citrullinated histone H4, and *Lm* (original magnification, x200).
(TIF)

**S8 Fig. Wt mice were infected i.v. with $2x10^5$ cfu of *Lm* EGDe. A) Serum levels of Sdc1 and Sdc4 ectodomains were measured at the indicated times post-infection (n = 3).** B) Sdc1 levels in liver, spleen, and lung urea extracts were measured before (control) and at 6 h pi (n = 3). C) Liver sections of Wt mice before (control) and at 24 h pi were immunostained for Sdc1 (original magnification, x200). D) *Sdc1*-/- mice were infected i.v. with $4.5x10^5$ cfu of *Lm* and injected with PBS or 500 U/mouse DNase I at 12 h pi and the liver bacterial burden was determined at 24 h pi (n = 6, *$p<0.05$).
(TIF)

**S9 Fig. *Lm* EGDe was diluted to the indicated OD600nm in BHI in the absence or presence of 20% PBS, 10 μg/ml HS, 10 or 20% mouse serum, or 20% serum with 10 μg/ml HS and incubated for 0, 2, or 10 h.** A) Bacterial aggregation was assessed by measuring turbidity at OD600nm (n = 3). B) Bacterial growth was measured by plating serial dilutions at 2 h and 10 h and counting *Lm* colonies (n = 3).
(TIF)

## Acknowledgments

We thank Dr. Darren Higgins (Harvard Medical School, Boston, MA) for the *Lm* strains and Dr. Gerald Pier (Harvard Medical School, Boston, MA) for the anti-PNAG antibodies.

## Author Contributions

**Conceptualization:** Rafael S. Aquino, Atsuko Hayashida, Pyong Woo Park.

**Formal analysis:** Rafael S. Aquino, Atsuko Hayashida, Pyong Woo Park.

**Funding acquisition:** Pyong Woo Park.

**Investigation:** Rafael S. Aquino, Atsuko Hayashida, Pyong Woo Park.

**Methodology:** Rafael S. Aquino, Atsuko Hayashida.

**Project administration:** Pyong Woo Park.

**Supervision:** Pyong Woo Park.

**Validation:** Rafael S. Aquino, Atsuko Hayashida.

**Visualization:** Rafael S. Aquino, Atsuko Hayashida.

**Writing – original draft:** Rafael S. Aquino, Atsuko Hayashida, Pyong Woo Park.

**Writing – review & editing:** Pyong Woo Park.

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
