## [Decision Letter · Decision Letter 0]

25 Mar 2020

Dear Dr. Park,

We are pleased to inform you that your manuscript 'Host Syndecan-1 Promotes Listeriosis by Inhibiting Intravascular Neutrophil Extracellular Traps' has been provisionally accepted for publication in PLOS Pathogens.

Best regards,

Timothy J. Mitchell, PhD

Associate Editor

PLOS Pathogens

Michael Otto

Section Editor

PLOS Pathogens

Kasturi Haldar

Editor-in-Chief

PLOS Pathogens

orcid.org/0000-0001-5065-158X

Michael Malim

Editor-in-Chief

PLOS Pathogens

orcid.org/0000-0002-7699-2064

This is an interesting and well described study on the role of syndecans in altering killing of Listeria by neurophil extracellar traps. The study could be improved by in vitro studies to expand on the mechanisms of how Listeria and host syndecan 1 interaction alters killing.

Reviewer Comments (if any, and for reference):

Reviewer's Responses to Questions

**Part I - Summary**

Reviewer #1: This study investigated the role of syndecan-1, an extracellular matrix compound, in Listeria monocytogenes infection. The authors find that L. monocytogenes exploits host syndecan-1 to spread in host tissues by inhibiting intravascular neutrophil extracellular traps. The study utilised genetically modified mouse strains, cell assays, and iimmunohistochemistry to test their hypothesis. The experiments were done in logical order, and the results described well.

Reviewer #2: (No Response)

Reviewer #3: This is a fascinating study in which mice lacking syndecan-1 have a surprising phenotype of enhanced ressitance to experimental Listeria infection. Through a progressive series of experimental manipulations the authors come to the conclusion that the syndecan mutant mice have enhanced formation of neutrophil extracellular traps in the liver that mediate clearance of Lm bacteria. The experimental evidence is logically presented and interpreted and is generally supportive of the authors' conclusions. Additional experiments that would help would include work to address mechanism(s) for Lm killing within NET and how syndecan reduces that.

**Part II – Major Issues: Key Experiments Required for Acceptance**

Reviewer #1: 1. The study utilised mouse model, cell assays and immunohistochemistry. While the authors tested their hypothesis successfully in these assays, the clinical relevance of their findings for listeria infections in humans is not known.

2. Dr Park's group previously reported that several other bacteria including Pseudomonas aeruginosa, Staphylococcus aureus, and Streptococcus pneumoniae also subvert the syndecan family of cell surface heparan sulfate proteoglycans to enhance their virulence in vivo. It is not clear how this study is different than the previous studies. Is subversion of the syndecan family of cell surface heparan sulfate proteoglycans by bacteria a universal strategy? If this study differs from the previous ones, then the authors clearly highlight this.

3. It is not clear how L. monocytogenes interfere with syndecan-1. Do the authors know which L. monocytogenes protein is utilised for syndecan-1 shedding?

4. While the wild type mouse and the syndecan deficient mice do not display differences in few selected parameters, the authors need to determine whether the impact of

syndecan-1 mutation effect the nutritional millue of host tissues and if this has any effect on L. monocytogenes transcriptome/proteome. It is well known that syndecan-1 has a role in binding and regulating the activity of various ligands including matrix components, growth factors, cytokines and morphogens.

Reviewer #2: This is an excellent study demonstrating how a pathogen hijacks the biology of a host factor to aid its survival resulting in enhanced virulence. The study is carefully designed and a number of controls and alternative explanations are considered. The experimental data presented are convincing and justify the conclusions. This manuscript, which is of a high level of interest to the readership of PLOS Pathogen, is generally clearly written.

Reviewer #3: Ideally it would be good to have a better idea for the basis for Lm susceptibliity to killing within NET and how syndecan impacts that. There are simple in vitro assays using ex vivo phagocytes which could help in that effort.

**Part III – Minor Issues: Editorial and Data Presentation Modifications**

Reviewer #1: (No Response)

Reviewer #2: Some comments/suggestions of editorial nature:

1) In the last paragraph of the Introduction section the authors write "Sdc1 ablation causes a gain of functioning both intragastric (i.g.) and intravenous(i.v.) mouse models……." . Please clarify if this statement refers to the experimental data presented in this manuscript.

2) It would be easier to read the Y-axis in the figures if the authors consistently used exponentials rather the the many zeros.

3) The quality of the fluorescence figures (particularly Fig 2) is poor.

Reviewer #3: (No Response)

PLOS authors have the option to publish the peer review history of their article (what does this mean?). If published, this will include your full peer review and any attached files.

Reviewer #1: No

Reviewer #2: No

Reviewer #3: No

---

## [Editor Report · Acceptance letter]

15 May 2020

Dear Dr. Park,

We are delighted to inform you that your manuscript, "Host Syndecan-1 Promotes Listeriosis by Inhibiting Intravascular Neutrophil Extracellular Traps," has been formally accepted for publication in PLOS Pathogens.

Best regards,

Kasturi Haldar

Editor-in-Chief

PLOS Pathogens

orcid.org/0000-0001-5065-158X

Michael Malim

Editor-in-Chief

PLOS Pathogens

orcid.org/0000-0002-7699-2064